# DEQ-MPC: DEEP EQUILIBRIUM MODEL PREDICTIVE CONTROL

## ABSTRACT

Incorporating task-specific priors within a policy or network architecture is crucial for enhancing safety and improving representation and generalization in robotic control problems. Differentiable Model Predictive Control (MPC) layers have proven effective for embedding these priors, such as constraints and cost functions, directly within the architecture, enabling end-to-end training. However, current methods often treat the solver and the neural network as separate, independent entities, leading to suboptimal integration. In this work, we propose a novel approach that co-develops the solver and architecture unifying the optimization solver and network inference problems. Specifically, we formulate this as a *joint fixed-point problem* over the coupled network outputs and necessary conditions of the optimization problem. We solve this problem in an iterative manner where we alternate between network forward passes and optimization iterations. Through extensive ablations in various robotic control tasks, we demonstrate that our approach results in richer representations and more stable training, while naturally accommodating warm starting, a key requirement for MPC.

## 1 INTRODUCTION

Incorporating task-specific priors within the policy training pipeline is often beneficial when solving robotic control problems. These priors, which often take the form of auxiliary constraints or cost functions, give the system designer an additional degree of control and flexibility while designing the system and play a vital role in enhancing safety, improving representation, and boosting generalization. Previous approaches to policy learning have explored various methods to embed such priors, including reward/loss shaping (Gupta et al., 2022), incorporating constrained optimization layers within the policy inference pipeline (Amos et al., 2018; Agrawal et al., 2020), adding parallel/post-hoc safety checks/controllers (Ames et al., 2019), adversarial training (Schott et al., 2024), and domain randomization (Chen et al., 2021).

Differentiable Model Predictive Control (MPC) layers (Amos et al., 2018) have emerged as a promising approach (Shrestha et al., 2023; Xiao et al., 2022; Diehl et al., 2023b). This method integrates MPC as a differentiable layer within neural network architectures, allowing for the embedding of constraints and cost functions directly into the network architecture, while enabling true end-to-end training of control policies. Importantly, they allow us to preserve the interpretability and safety guarantees associated with traditional MPC while providing a general framework applicable to a diverse range of robotic control problems. Furthermore, it allows for test-time modifications of the MPC problem and facilitates online adaptation, offering increased flexibility and generalizability – a critical feature in dynamic environments.

While offering several advantages, standard differentiable MPC layers often treat the optimization solver as a black-box differentiable layer within the neural network (NN) architecture. This simplification, while convenient, overlooks the unique characteristics of MPC solvers that set them apart from typical NN layers. MPC solvers are implicit layers and hence inherently iterative as opposed to typical explicit layers. They often suffer from ill-conditioning, non-convexities and discontinuities, potentially leading to unstable training dynamics. Additionally, MPC solvers frequently possess specialized structures that enable efficient warm-starting – a valuable property in recurrent control scenarios that is not fully leveraged in differentiable MPC frameworks.

To address these limitations, we propose Deep Equilibrium Model Predictive Control (DEQ-MPC), a novel approach that unifies the optimization solver and the neural network architecture. Instead

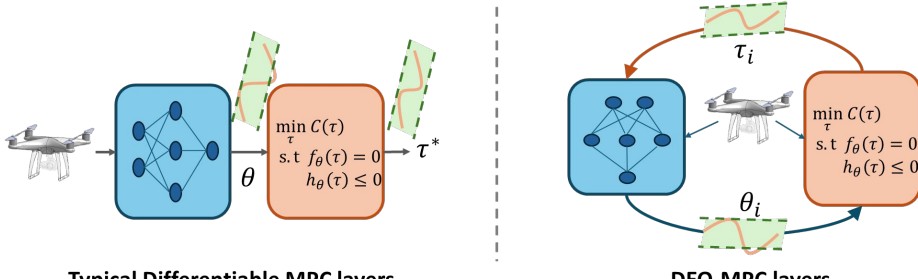

**Typical Differentiable MPC layers**  **DEQ-MPC layers**

Figure 1: We propose DEQ-MPC layers (right) as a direct improvement over differentiable MPC layers (left). These layers offer increased representational power, smoother gradients, and are more amenable to warm-starting. DEQ-MPC layers formulate MPC parameter estimation ($\theta$) and trajectory optimization ($\tau$) as a joint fixed-point problem, solving them in an alternating iterative manner, instead of the single-shot sequential inference used in differentiable MPC setups. This approach allows the network to adapt the optimization parameter estimates, $\theta_i$, based on the current optimizer state, $\tau_i$, enabling a richer feedback process. The specific example in the figure shows a trajectory tracking example, where the robot observations (quadrotor) are fed to the system. The network predicts the waypoints $\theta_i$ (optimization parameters). The solver solves the tracking problem to spit out solved trajectories $\tau_i$ to track the waypoints $\theta_i$.

of treating the optimization layer as just another layer within the network, we formulate a joint inference and optimization problem as shown in figure 1, where we treat the network inference and the optimization problem as a unified system and jointly compute a fixed point over them. Thus, the network outputs can now depend on the solver iterates and vice-versa, thereby, allowing a tight coupling between the two. The fixed point is computed by alternating between the network forward pass (conditioned on the most recent optimizer iterate) and the optimization solver iterations (conditioned on the most recent network outputs) until the joint system reaches an equilibrium (hence the name DEQ-MPC, i.e, Deep Equilibrium Model Predictive Control).

This joint inference/optimization framework also allows us to explore several interesting aspects of the solver and architecture design. Specifically, for the optimization solver, we implement an augmented Lagrangian (AL) solver which works well with warm-starting and is robust at handling arbitrary non-linear constraints. This is important for the joint fixed point process as it allows us to change the optimization parameters (i.e, network outputs, $\theta_i$) between successive optimization iterates. For the architecture, we experiment with parameterizing the network architecture itself as a Deep Equilibrium model (DEQ), a type of implicit neural network that computes the outputs/latents as a fixed point of a non-linear transformation. It can be seen as an infinite depth network which applies the same layer an infinite number of times eventually reaching a fixed point in the outputs/latents. This iterative fixed point finding procedure blends nicely with the equilibrium/fixed point finding nature of the overall system. We observe nicer stability properties when using a DEQ as the network architecture when going to more complicated settings.

This unified approach results in several key benefits: First, it enables richer representations by allowing the network to adapt its features/outputs depending on the solver state. Second, it allows us to naturally compute smoother gradients during training, facilitating more stable and efficient learning. Third, it inherently accommodates warm-starting, leveraging the recurrent nature of MPC to improve computational efficiency and solution quality. DEQ-MPC thus offers a more robust and flexible framework for integrating optimization-based control with deep learning.

The primary contributions of this work are as follows: (1) We introduce DEQ-MPC, a novel framework that seamlessly integrates MPC layers into deep networks. (2) Through extensive ablation studies, we show that this unified approach results in richer representations, improved gradient flow, and enhanced suitability for warm-starting, compared to standard differentiable MPC methods. (3) We propose a training setup specifically for streaming MPC applications that leverages warm-starting across time steps. (4) We provide empirical evidence demonstrating the advantages of DEQ-MPC on trajectory prediction and tracking problems across various continuous control tasks that require strict constraint satisfaction. While this paper focuses on MPC to ground our methods in a concrete context, we believe that the insights and techniques developed here have broader implications for integrating constrained optimization layers into deep networks in a wide range of applications.

## 2 RELATED WORK

Differentiable optimization layers were introduced as a means to embed convex optimization problems (Amos & Kolter, 2017; Agrawal et al., 2019) as differentiable units within deep networks. Recent works have extended the range of optimization and fixed-point problems that can be made differentiable (Gould et al., 2021; Landry et al., 2019; Jin et al., 2020; Pineda et al., 2022). Since their introduction, they have been applied to a variety of robotics problems, such as state estimation (Yi et al., 2021), SLAM (Teed et al., 2023; Lipson et al., 2022), motion planning (Bhardwaj et al., 2020; Landry et al., 2019) and control (Amos et al., 2018; Agrawal et al., 2020) for applications such as autonomous driving (Shrestha et al., 2023; Diehl et al., 2023a), navigation (Xiao et al., 2022; Diehl et al., 2023b), and manipulation (Landry et al., 2019). We specifically look at differentiable Model Predictive Control problems building off of work such as (Amos et al., 2018; Agrawal et al., 2020), which show how to differentiate through simple trajectory optimization problems and use them as layers within broader differentiable pipelines for tasks such as behavior cloning and system identification. Various follow-up works have demonstrated that policies with differentiable optimization layers can be applied more broadly within model-free (Romero et al., 2023) and model-based RL (Wan et al., 2024) pipelines as well. A separate line of work has also explored extending to more complicated trajectory optimization problems differentiable such as problems with cone constraints (Howell et al., 2022), general non-linear programs (Landry et al., 2019).

However, incorporating MPC layers (and optimization layers) within deep networks as just another black box differentiable layer can often come with its own set of challenges. The bi-level problem can often be very non-convex resulting in the local gradient direction being mis-aligned with the desired global update direction (Landry et al., 2019; Amos et al., 2018). Likewise, the gradient landscape often has discontinuities resulting in undesirable gradient artifacts (Suh et al., 2022; Antonova et al., 2023). Furthermore, the problem structure can also often result in very high variance in gradients (Gurumurthy et al., 2024). It's often challenging to incorporate warm-starting techniques as the problem parameters change with each problem instance (Sambharya et al., 2024) resulting in long inference and solve times. The network predicted constraint parameters can often be infeasible (Donti et al., 2021), resulting in undefined problem solutions or gradients. Some modelling assumptions in the optimization layer are often not faithful to the real data causing model mismatch problems (Gurumurthy et al., 2023a). Tackling these challenges is critical to making the use of optimization layers in modern deep learning pipelines more practical.

Our proposed method to formulate the network inference and the optimization problem as a joint equilibrium finding problem seeks to address some of these issues such as representation ability, gradient discontinuities and suitability for warm-starting. Previous works have also used similar motivations to pose the inference and the optimization problem as a joint equilibrium finding problem albeit for problems without any constraints. Gurumurthy et al. (2021) pose the task of latent variable optimization to solve inverse problems as a joint inference and optimization problem over the network outputs and the latent variables. But they primarily look at simple least squares problems. Likewise, Teed & Deng (2021); Teed et al. (2023); Lipson et al. (2022) use a similar framework in the context of SLAM/visual odometry/object pose estimation problems where they specifically look at non-linear least squares, bundle adjustment problems. Our paper generalizes these methods to general constrained optimization problems and grounds it in MPC problems. Furthermore, unlike previous work, we directly compare with the vanilla differentiable optimization alternative and tease apart the specific advantages offered by the joint optimization approach.

## 3 BACKGROUND

### 3.1 DIFFERENTIABLE MODEL PREDICTIVE CONTROL

Model Predictive Control (MPC) solves a finite-horizon optimal control problem at each time step. The general form of an MPC problem can be expressed as:

$$\tau^*_{0:T} = \arg\min_{\tau_{0:T}} \quad \sum_t C_{\theta,t}(\tau_t)$$
$$\text{subject to} \quad x_0 = x_{\text{init}}, \ x_{t+1} = f_\theta, \ h_\theta(\tau_t) \leq 0, \ t = 0, \dots, T, \tag{1}$$

where $\tau_t = (x_t, u_t)$ represents the state-action pair, $C_{\theta,t}$ is the cost function, $f_\theta$ is the dynamics function and $h_\theta$ are some inequality constraints on the trajectory (e.g. safety constraints, joint lim-

its, etc.). This non-linear optimization problem is typically solved using non-linear programming techniques.

The key innovation in differentiable MPC is the computation of gradients with respect to the problem parameters. At the solution, gradients are computed using the implicit function theorem. Specifically, let $z^* = (\tau^*, \lambda^*, \nu^*)$ be the primal-dual solution to the KKT conditions $F(z, \theta) = 0$ of equation 1. The gradient of the solution with respect to the parameters $\theta$ can be computed as:

$$\frac{\partial z^*}{\partial \theta} = -\left(\frac{\partial F}{\partial z}\right)^{-1} \cdot \frac{\partial F}{\partial \theta}, \tag{2}$$

where $\frac{\partial F}{\partial z}$ is the KKT matrix at the solution. This approach allows the MPC solver to be integrated into end-to-end learning pipelines, enabling the incorporation of domain knowledge and constraints directly into learned control policies.

## 3.2 DEEP EQUILIBRIUM MODELS

Deep Equilibrium Models (Bai et al., 2019) are a class of implicit deep learning models that compute the output as a solution to a fixed point problem. Specifically, given an input $x \in \mathcal{X}$, computing the forward pass in a DEQ model involves finding a fixed point $z \in \mathcal{Z}$, such that

$$z^\star = d_\phi(z^\star, x), \tag{3}$$

where, $d_\phi : \mathcal{Z} \times \mathcal{X} \to \mathcal{Z}$ is some parameterized layer conditioned on input $x$, $\mathcal{Z}$ denotes the hidden state or outputs of the network which we are computing the fixed point on, $\mathcal{X}$ denotes the input space, and $\phi$ denotes the parameters of the layer. Computing this fixed point (under proper stability conditions) corresponds to the "infinite depth" limit of repeatedly applying the function $z^+ := d_\phi(z, x)$ starting at some arbitrary initial value of $z$ (typically 0).

## 4 METHOD

In this section, we introduce DEQ-MPC and detail key design decisions in network architecture, solver, and gradient computations, along with their benefits.

### 4.1 PROBLEM GROUNDING THROUGH TRAJECTORY PREDICTION AND TRACKING

We begin by grounding our discussion in a simple trajectory prediction and tracking example. This example helps make the following discussion more intuitive and motivates our design decisions. Additionally, it will serve as the default configuration for all our subsequent experiments.

Consider a system with dynamics $f$. Given a dataset of optimal trajectories across different initial and environmental conditions, we seek to learn a policy that solves the imitation learning problem while respecting several constraints. We model this policy as consisting of two components. The first is a neural network $\text{NN}_\phi$ that predicts the waypoints $\theta_{0:T}$ to be tracked for the next $T$ time steps given the current state $x_\text{init}$ and some observations $o$:

$$\theta_{0:T} = \text{NN}_\phi(x_\text{init}, o). \tag{4}$$

The second is an MPC solver that solves the trajectory tracking problem to compute dynamically feasible trajectories $\tau_{0:T}$ that track the waypoints while satisfying the required constraints:

$$\tau_{0:T}^* = \arg\min_{\tau_{0:T}} \quad \sum_t \|x_t - \theta_t\|_Q^2 + \|u_t\|_R^2$$
$$\text{subject to} \quad x_0 = x_\text{init}, \ x_{t+1} = f_\theta(\tau_t), \ h_\theta(\tau_t) \leq 0, \ t = 0, \ldots, T. \tag{5}$$

In a standard differentiable-MPC setup these two components are executed sequentially, one after the other as shown in figure 1. The outputs of the system, $\tau_{0:T}^*$ are used to compute a loss, $\ell(\tau_{0:T}^*)$, such as a supervised L1 loss with some expert trajectory demonstrations $\tau_{0:T}^\text{exp}$. The loss is then optimized using a stochastic gradient optimizer to learn the network parameters.

## 4.2 DEQ-MPC

### 4.2.1 THE INFERENCE PROBLEM, ARCHITECTURE AND SOLVER

MPC solvers are implicit layers and hence inherently iterative. Using a single parameter estimate throughout the solver iterations is inefficient and potentially ineffective, especially for non-linear

optimization problems. To address this, DEQ-MPC modifies the single-shot inference problem described in equations 4 and 5 into a joint inference/optimization problem over the network outputs and the optimizer iterates. This approach, illustrated in Figure 1, allows us to condition the network outputs (optimization parameters, $\theta$) on the optimizer state $\tau$ and vice versa. This can be expressed as a single constrained optimization problem:

$$\tau_{0:T}^*, \theta^* = \underset{\tau_{0:T}, \theta}{\arg\min} \quad \sum_t C_{\theta, t}(\tau_t) \tag{6}$$

$$\text{subject to} \quad x_0 = x_{\text{init}}, \ x_{t+1} = f_\theta, \ h_\theta(\tau_t) \leq 0, \tag{7}$$

$$\theta = \text{NN}_\phi(x_{\text{init}}, o, \tau_{0:T}), \ t = 0, \ldots, T, \tag{8}$$

where the last constraint expresses the neural network inference as an equality constraint. This simply represents a large non-linear optimization problem which can be potentially be solved in several ways. However, typical non-linear optimization solvers struggle with having neural network layers as constraints due to the nastiness of the resulting constraint Jacobians. We propose to solve this problem using the alternating direction method of multipliers (ADMM) algorithm (Boyd et al., 2011), alternating between (1) solving the MPC optimization problem (with fixed $\theta$), equations 6 and 7 using the augmented Lagrangian (AL) method and (2) the constraint projection step, equation 8 (i.e, the standard neural net inference to compute $\theta$ with fixed $\tau$). Specifically, we alternate between the following two operations for $N$ iterations or until convergence,

$$\theta^i = \text{NN}_\phi(x_{\text{init}}, o, \tau^{i-1}), \tag{9}$$

$$\tau^i = \text{MPC-m}_{\theta^i}(x_{\text{init}}, \tau^{i-1}), \tag{10}$$

where MPC-m performs $m$ solver iterations using the AL algorithm, with the most recent parameter estimate $\theta^i$ from the network and warm-started using $\tau^{i-1}$ from the last MPC-m solve. The initial value $\tau^0$ are initialized at $x_{\text{init}}$ and zero controls across time steps. We refer to each alternating step as a DEQ-MPC-iteration, with the super-script, $i$, denoting the iteration count. This is illustrated in figure 1. This iterative inference/optimization approach enables the network to provide an initial coarse parameter estimate and iteratively refine it based on the solver's progress.

*Choice of N, m*: Empirically, we find that updating the MPC parameters $\theta$ every two AL iterations ($m = 2$) is sufficient to obtain most of the gains. Furthermore, DEQ-MPC typically converges within $N = 6$ DEQ-MPC-iterations with $m = 2$ and thus we use these values for all our experiments. We discuss the considerations around the convergence of this alternating problem in section A.5.

**Network architecture.** We explore two architectural choices for $\text{NN}_\phi$ with distinct trade-offs:

(1) *DEQ-MPC-NN:* We implement $\text{NN}_\phi$ as a standard feedforward network. While this proves to be a simple and effective choice for most scenarios, it has limitations. The iterative nature of the DEQ-MPC framework can lead to instabilities when using a generic feedforward architecture, particularly in complex settings. Moreover, this architecture is somewhat computationally inefficient, as it doesn't leverage the similarity of computations across successive iterations – each iteration starts anew without reusing previous computational results.

(2) *DEQ-MPC-DEQ:* To address these limitations, we also implement $\text{NN}_\phi$ architecture itself as a DEQ network (Bai et al., 2019). Specifically, the network inference step in equation 9 is itself expressed as a fixed point finding problem:

$$z_i^\star = d_\phi(z_i^\star, x_{\text{init}}, o, \tau_{i-1}). \tag{11}$$

This computation yields the pre-final layer network latent state, as described in section 3.2. The updated MPC parameters are then obtained through $\theta_i = g_\phi(z_i^\star)$. Note that this fixed point solve is distinct from the equilibrium computations in the DEQ-MPC-iterations discussed earlier. The fixed point iteration discussed here is simply computing the network inference (i.e constraint projections) from equation 9 when using a DEQ network. Furthermore, given that we expect these fixed points across successive DEQ-MPC-iterations to be similar, we can also warm-start these fixed point iterations, i.e, $z_i$ can be conveniently initialized with $z_{i-1}$ while computing the fixed points. This allows us to re-use the network computation from earlier iterations. We use a standard fixed point solver (Walker & Ni, 2011) to compute this fixed point.

**MPC-m solver.** We use the AL algorithm (Nocedal & Wright, 2006; Toussaint, 2014) for the MPC solver. This is motivated by its ability to accommodate arbitrary non-linear constraints as

penalties and its suitability for warm-starting. The penalty-based approach also allows us to use the unconverged iterations as smoothed/relaxed versions of the problem to handle discontinuities (more discussion in section 4.2.2). Our solver implementation is friendly with both CPU and GPU.

Specifically, for the general MPC problem in equation 1, we form the following Lagrangian

$$\mathcal{L}(\tau, \lambda, \eta, \mu) = \sum_t C_{\theta,t}(\tau_t) + \lambda^T h_\theta(\tau) + \eta^T k_\theta(\tau_t, x_{t+1}) + \frac{\mu}{2} \|h_\theta(\tau_t)^+\|_2^2 + \frac{\mu}{2} \|k_\theta(\tau_t, x_{t+1})\|_2^2, \quad (12)$$

where $h_\theta(\tau_t) \leq 0$ are the inequality constraints and $k_\theta(\tau_t, x_{t+1}) = 0$ are all the equality constraints (including the dynamics and initial state constraints), $\lambda$ and $\eta$ are the corresponding Lagrange multipliers and $\mu > 0$ is the penalty parameter. $h_\theta(\tau_t)^+$ represents an element-wise clipping at zero $\max(0, h_\theta(\tau_t))$. The AL method alternates between the updates of the primal variables, dual variables and penalty parameters until convergence as described in algorithm 1 in the appendix.

However, with MPC-m, we only perform $m$ AL iterations. Furthermore, we implement warm-starting across DEQ-MPC iterations: all the variables $(\tau^i, \lambda^i, \eta^i, \mu^i)$ at the $i$-th DEQ-MPC-iteration are initialized with the corresponding values computed at the end of the $(i-1)$-th DEQ-MPC-iteration, $(\tau^{i-1}, \lambda^{i-1}, \eta^{i-1}, \mu^{i-1})$.

### 4.2.2 LOSS AND GRADIENTS

In this section, we address the challenges of gradient computation when differentiating through an augmented Lagrangian solver by modifying the gradient and loss computation. We discuss the details of gradient computation for the DEQ network in the appendix A.2.

**Augmented Lagrangian gradients.** Previous work (Suh et al., 2022; Antonova et al., 2023) has shown that computing gradients through optimization problems can be problematic due to inherent discontinuities in the landscape and have proposed various relaxations to tackle this problems. We take inspiration from these approaches and propose a relaxation for use with our solver.

We compute the gradient through the AL solver using the implicit function theorem equation 2 where the function $F(\cdot)$ now represents the Lagrangian's gradient $\nabla_\tau \mathcal{L}_\theta(\tau, \lambda, \mu)$. Thus the IFT gradient is

$$\nabla_\theta \tau = -(\nabla_\tau^2 \mathcal{L})^{-1} \nabla_{\theta\tau} \mathcal{L} \quad (13)$$

$$= -(Q + \mu A^T A + \mu G^T G)^{-1} \nabla_{\theta\tau} \mathcal{L}. \quad (14)$$

where, $A$ and $G$ are the constraint Jacobians of the equality and inequality constraints respectively. At convergence, the value of $\mu$ is very high. This results in the components of the gradient in the column space of the linearized active constraints getting squished to zero. Thus, when the constraints are non-linear/discontinuous, and the optimizer converges to some arbitrary active sets, the gradients computed using equation 13 are also arbitrary/meaningless.

We instead propose to compute losses on multiple unconverged intermediate iterates along the optimizer iterations and minimize all of them during training. Thus, the gradients at the initial optimization iterates are computed with smaller values of the penalty parameter $\mu$ while the latter ones are computed with larger values of $\mu$. As a result, the earlier optimization iterates obtain relaxed gradients even when the optimizer converges to arbitrary active sets, while the latter iterates obtain "accurate" gradients as long as the optimizer converges to the "right" active sets. This provides a *natural curriculum*, where the initial iterates converge to smoothed/relaxed solutions and the latter iterates are then incentiviced to nail down the details.

**Losses.** We primarily look at the imitation learning problem and thus use a simple supervised learning objective. We use an L1 loss over the output states against the corresponding ground truths for supervision. As discussed before, we compute losses on multiple intermediate iterates and back-propagate gradients through all of them. The resulting objective for a single instance is

$$\ell(x_{0:T}^{\exp}, x_{0:T}^{1:I}) = \sum_{t=0:T} \sum_{j=1:I} \|x_t^{\exp} - x_t^i\|_1, \quad (15)$$

where $x_{0:T}^{\exp}$ are expert demonstrations and $x_{0:T}^{1:I}$ are the states output by the model across $I$ iterations.

### 4.2.3 WARM-STARTING AND STREAMING

**Warm-starting.** MPC problems, like various other optimization problems, benefit from warm-starting (Howell et al., 2019; Le Cleac'h et al., 2024). The resulting speedups from warm-starting are often critical for real-world deployment (Nguyen et al., 2024). In the context of MPC, this involves reusing the converged MPC iterate from the previous time step as initialization for the current solve, so as to minimize the number of optimizer iterations needed at each time-step. Specifically, the MPC problem solving for $\tau_{t:T+t}$ at time, $t$, is warm started with the final solution computed at the previous time-step $\hat{\tau}_{t-1:T+t-1}$. The initialization for $\tau_{t:T+t}$ is thus computed by concatenating $\tau_{t:T+t} = [\hat{\tau}_{t:T+t-1}, \hat{\tau}_{T+t-1}]$, where $\hat{\tau}_{T+t-1}$ is assumed to be a reasonable estimate for $\tau_{T+t}$.

The augmented Lagrangian algorithm provides a very convenient way for incorporating the warm-started initialization. We simply initialize $\tau$ with the warm-starting estimate, reset the dual variables $\lambda$ and $\eta$ to zeros and set the initial value of $\rho = \rho_{\max}/10^{N*m-i}$ where $(N * m - i)$ is the total number of AL iterations we expect to perform after warm-starting. In standard differentiable-MPC setups, the network infers the MPC parameters afresh at each successive time step. These parameter estimates can often be arbitrarily far from the previous estimates, thus requiring a significant number of AL iterations post warm-start. On the other hand, in DEQ-MPC, the network is conditioned on the previous optimizer iterate. This allows us to train the network to predict consistent parameter estimates across time-steps by training it specifically for the streaming setting as described below.

**Streaming training.** We customize the training procedure to suit the warm-started streaming setup. Given a sampled ground truth trajectory $\tau_{0:T+L}^{\exp}$, we break the inference problem into a two step process. First, we solve for $\tau_{0:T}$ given $x_0^{\exp}$ as usual without any warm-starting. Then, we successively solve $L$ problems for $\tau_{t:T+t}$ for $t = 1 \ldots N$ with the iterates warm-started with solution from the previous solve, $\tau_{t-1:T+t-1}$. Then we simply compute losses on all the intermediate optimization iterates (from both steps) and supervise them using the corresponding ground truths as described in section 4.2.2. For all of our experiments we use $L = 2$.

## 5 EXPERIMENTS

We demonstrate the effectiveness of our proposed modifications across a variety of systems. Additionally, we present ablation studies to highlight the specific advantages of DEQ-MPC.

**Setup.** We use the trajectory prediction and tracking problem, discussed in section 4.2.1, as our default experimental setting. For each task, we generate ground truth trajectories using 'expert' policies trained with a state-of-the-art on-policy reinforcement learning algorithm (Gurumurthy et al., 2023b). We partition the generated data into training (90%) and validation (10%) sets. Models are trained via supervised learning to predict the next T steps in a trajectory, given the current state as input, as outlined in section 4.2.1. By default, $T = 5$ for all environments unless otherwise specified.

We evaluate the models in two ways. First, when evaluating their effectiveness as a generic optimization layer within differentiable pipelines, we compare the models based on their validation errors. Second, to evaluate their suitability for the MPC setting, we implement them as feedback policies in the original environment using a receding horizon approach and compute the average returns over 200 rollouts.

**Variants/Baselines.** Throughout the experiments and ablations, we compare our methods (DEQ-MPC-*) against their corresponding differentiable MPC counterparts (Diff-MPC-*):

*DEQ-MPC-DEQ:* Our method where the network architecture uses a DEQ model.

*DEQ-MPC-NN:* Our method where the network architecture uses a standard feed forward network.

*Diff-MPC-NN:* A standard differentiable-MPC setup where a standard feedforward network predicts the MPC problem parameters (waypoints) in one shot, which are then used to solve the MPC problem. The loss is computed at the converged iterate and backpropagated using IFT.

*Diff-MPC-DEQ:* This uses the same setup as Diff-MPC-NN except that the network architecture is replaced with a DEQ.

**Network architecture.** The trajectory prediction and tracking problem is inherently sequential, as the network takes the current system state as input and predicts the future $T$ states to be tracked.

Given this sequential nature, we employ a temporal convolution-based architecture for both the DEQ and the feedforward network used in our experiments. Additional details regarding the architecture of both the NN and DEQ are provided in appendix A.4.

## 5.1 COMPARISON RESULTS

We evaluate the methods on a series of underactuated continuous control tasks with constraints:

*Pendulum*: We consider a standard pendulum swing-up task with imposed control limits of $\pm 5$ units, which are modeled as inequality constraints in the MPC problem. The system has a state dimension of 2 and a control dimension of 1. The dataset consists of 300 trajectories.

*Cartpole*: We consider a standard cartpole swing-up task with imposed control limits of $\pm 100$ units, which are modeled as inequality constraints in the MPC problem. The system has a state dimension of 4 and a control dimension of 1. The dataset consists of 300 trajectories.

*Quadrotor*: We use the Quadrotor model from (Jackson et al., 2022) where the objective is to guide the quadrotor from a randomly initialized position to the origin. In this task, we impose control limits on all motors, constrained to the range $[11.5, 18.3]$ units, which are formulated as inequality constraints in the MPC problem. The system has a state dimension of 12 and a control dimension of 4. The dataset consists of 2000 trajectories.

*QPole*: We attach a free-rotating pole to the center of mass (COM) of the *Quadrotor* while maintaining the same control authority. The task is to guide the quadrotor to the origin while ensuring the pole is swung up, which is very dynamic and challenging. The system has a state dimension of 14 and a control dimension of 4. The dataset consists of 2000 trajectories.

*QPoleObs*: To increase complexity of the *QPole* task, we introduces 40 obstacles. The quadrotor aims to reach the origin with the pole swung up, while maintaining a minimum distance $r$ from the obstacles at all points. By default, $r = 0.5$ units unless specified otherwise. The collision avoidance constraints are modeled in the MPC layer as $(\|x_d - x_0\|_2^2 \geq r^2)$, where $x_d$ represents the COM position of the drone and $x_0$ is the center of the obstacle. The dataset consists of 2000 trajectories.

Across these experiments we use a prediction/planning horizon of $T = 5$ for the MPC. We run the policy for 200 time steps in a receding horizon fashion for evaluation and average the returns across 200 different runs. The policies are trained and executed in the streaming setting (section 4.2.3) with a single DEQ-MPC-iteration (DEQ-MPC variants)/two AL iterations (Diff-MPC variants) with warm-starting across environments, except in the QPoleObs env, where all methods needed two DEQ-MPC-iterations (DEQ-MPC)/four AL iterations (Diff-MPC). (Note that each DEQ-MPC iteration itself also does exactly two AL iterations with $m = 2$). Table 1 shows the normalized returns obtained by each policy for each task averaged across policies trained with three random seeds/dataset splits. The returns presented in the table are normalized such that the returns of the expert policy are 1.00. We observe that the DEQ-MPC variants consistently perform better than the Diff-MPC counterparts across most environments. While DEQ-MPC-DEQ performs consistently well across all environments, we observed that DEQ-MPC-NN occasionally got unstable (e.g. resulting in its sub-par performance in the Cartpole balancing task).

Table 1: Performance comparison across various environments, with values normalized against the expert return for each environment. A higher score indicates better performance.

| Environment | Pendulum | Cartpole | Quadrotor | QPole | QPoleObs |
|---|---|---|---|---|---|
| Diff-MPC-NN | 0.77 (±0.04) | 0.93 (±0.05) | 0.96 (±0.02) | 0.76 (±0.05) | 0.83 (±0.02) |
| Diff-MPC-DEQ | 0.78 (±0.04) | 0.97 (±0.06) | 0.88 (±0.01) | 0.72 (±0.03) | 0.71 (±0.05) |
| DEQ-MPC-NN | **0.94 (±0.02)** | **1.00 (±0.09)** | **1.00 (±0.01)** | **0.87 (±0.02)** | **0.94 (±0.03)** |
| DEQ-MPC-DEQ | **0.94 (±0.04)** | 1.13 (±0.01) | 0.98 (±0.01) | 0.85 (±0.03) | 0.90 (±0.03) |

## 5.2 ABLATIONS

We explore three aspects of the model: representation, training stability and warm-startability. We perform all the experiments with the QPole environment unless otherwise specified.

### 5.2.1 REPRESENTATION ABLATIONS

We present three ablations to demonstrate the representation capabilities of DEQ-MPC. First, we demonstrate that DEQ-MPC variants scale more effectively with both model capacity and dataset size. Second, we show that DEQ-MPC variants experience less performance degradation than Diff-MPC variants as constraint complexity increases. Additional ablation experiments in appendix A.1 investigate (1) the impact of varying the horizon length of the problem and (2) the impact of the optimization parameter updates in DEQ-MPC vs Diff-MPC.

**Generalization.** Figure 2 shows the validation error as we vary the training set size from $0.2$ to $1.0\times$ of the full training set. We observe that the representational benefits of the DEQ-MPC models are evident even with smaller datasets. Additionally, we observe clear signs of saturation in the performance of the Diff-MPC variants as the dataset size increases, whereas, the performance of the DEQ-MPC variants continue to improve with increasing dataset size, suggesting that the higher representation power translates into an increased ability to ingest larger datasets. We also plot the validation scores when training the networks (DEQ and NN) directly with supervised learning, i.e, without any MPC layers and observe that the models themselves also tend to saturate with increasing capacity indicating that the benefits indeed arise from interleaving the network and the solver.

**Network capacity.** Figure 3 shows the validation error as we vary the network hidden state size from 128 to 1024 for all the models. We observe that the DEQ-MPC variants benefit more from the higher network capacity than the Diff-MPC variants. Infact, the DiffMPC variants saturate beyond hidden size of 512 whereas the DEQ-MPC variants continue to improve. This shows that the DEQ-MPC variants also have a better ability to utilize additional model capacity if available and thus are also more amenable to scaling.

**Constraint hardness.** We add 40 obstacles to the environment along with additional collision avoidance constraints represented as $(\|x_d - x_0\|_2^2 \geq r^2)$ to the MPC layer, where $x_d$ is the COM position of the drone and $x_0$ is the center of the obstacle. Figure 4 shows the returns obtained by different models on the task as we change the obstacle radius $r$ from 0.20 to 0.50. We observe that not only are the performance improvements of the DEQ-MPC variants preserved as we add additional constraints (as obstacles), but in fact the difference increases as the task gets harder and the obstacle sizes increase. Note that these are not warm started runs, i.e, we run the optimizer from scratch in order to decouple the effects of warm-starting from the representational effects.

### 5.2.2 TRAINING STABILITY

**Gradient niceness.** In this ablation study, we aim to illustrate the impact of naively applying IFT gradients computed for the AL solver during training. Figure 5 presents the validation errors during training for DEQ-MPC-DEQ (where we compute losses across multiple intermediate AL iterates and backpropagate) and Diff-MPC-DEQ (where gradients are computed only through

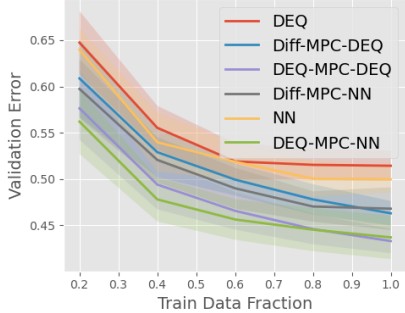

Figure 2: Generalization ablations

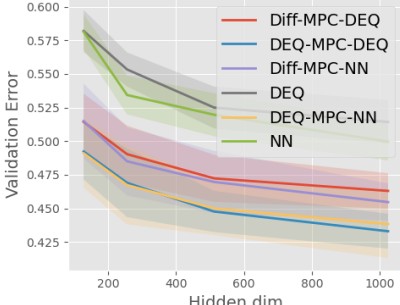

Figure 3: Network capacity ablations

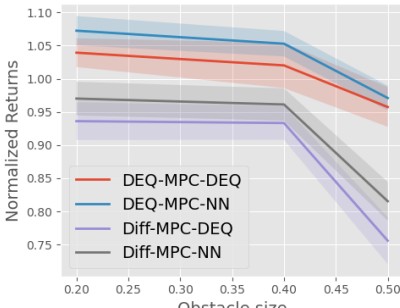

Figure 4: Constraints hardness

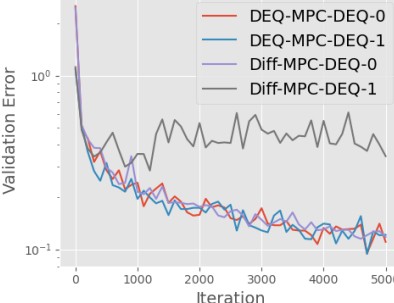

Figure 5: Gradient instability ablations

the final AL iterate). We observe significant instability in the training process for the latter model when tight control limits are enforced as inequality constraints (denoted with postfix 1). This leads to overall training instabilities. In contrast, training remains stable and smooth for both models when the inequality constraints are removed (denoted with postfix 0).

**MPC parameter sensitivity.** Figure 6 shows the validation errors of the models as we vary the velocity coefficients in the MPC cost matrix $Q$ (lower values lead to higher problem sensitivity). We observe that as $Q$ gets increasingly ill-conditioned, the sensitivity of the system increases. This results in the models becoming increasingly more unstable during training. The validation errors plotted represent the 'best' performance of the model throughout training (typically just before the training became unstable). We observe that DEQ-MPC-DEQ remains stable for the largest range of values. Even DEQ-MPC-NN, although best performing with well conditioned Q values, quickly gets very unstable as the conditioning worsens.

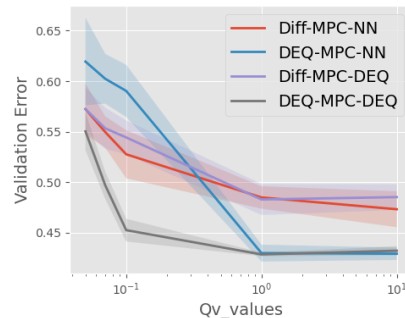

Figure 6: Cost parameter ablations

### 5.2.3 WARM-STARTING ABLATIONS

Figure 7 shows the returns obtained by models trained and evaluated with different number of DEQ-MPC/AL iterations in the streaming setup (with warm-starting) discussed in section 4.2.3. Note that, each DEQ-MPC iteration does exactly two AL iterations ($m = 2$). We set the number of streaming training steps, $L = 2$ for all experiments. We observe that the difference between the performance of the DEQ-MPC models and the corresponding differentiable MPC variants increase significantly as we reduce the number of warm-started AL iterations/DEQ-MPC iterations. DEQ-MPC models due to their iterative setup naturally adapt to the warm-started streaming setup, given that the warm-starting required at each new time-step is very similar to the warm-starting done in DEQ-MPC across DEQ-MPC iterations.

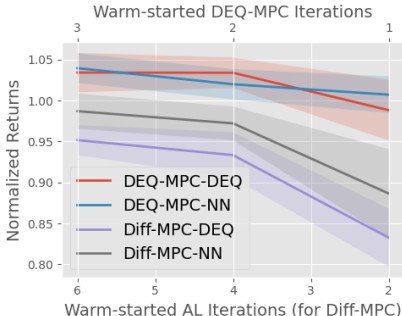

Figure 7: Warm-starting ablations

## 6 DISCUSSION AND FUTURE WORK

**Discussion.** Our experimental results highlight several key advantages of DEQ-MPC over differentiable MPC layers. The performance gap between DEQ-MPC variants and Diff-MPC becomes increasingly apparent as task complexity increases, whether through harder constraints, longer planning horizons, or increased problem sensitivity. A particularly promising aspect of DEQ-MPC is its favorable scaling behavior. Unlike Diff-MPC variants which show signs of performance saturation, DEQ-MPC models continue to improve with increasing dataset size and network capacity. This suggests potential for exploiting scaling laws in robotics applications. Furthermore, DEQ-MPC's effectiveness in warm-starting scenarios, requiring fewer augmented Lagrangian iterations while maintaining performance, offers significant practical advantages for real-world deployment. Interestingly, there exist trade-offs even between the DEQ-MPC variants. While DEQ-MPC-NN performs slightly better on average, DEQ-MPC-DEQ remains stable across a wider range of conditions compared to DEQ-MPC-NN, suggesting a trade-off between performance and stability.

**Limitations and future work.** Several important directions remain for future work. While our method is designed to be general, our current evaluation focuses primarily on trajectory tracking problems. Exploring the applicability of DEQ-MPC to a broader class of MPC and constrained optimization problems, both within and beyond robotics, would be valuable. Additionally, investigating whether the representational richness of DEQ-MPC can be leveraged effectively beyond the imitation learning setup such as in reinforcement learning settings to directly learn constrained optimal policies could be a promising line of future work. Finally, given the strong performance in constraint handling, exploring DEQ-MPC in safety-critical scenarios such as human-robot interaction settings with dynamic obstacles would be an interesting direction for future research.

# 7 REPRODUCIBILITY STATEMENT

We provide the code to reproduce the experiments in the supplementary material with all the details provided in the corresponding README.md file to train and evaluate the models. We also plan on releasing the code upon paper acceptance.

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

## A APPENDIX

### A.1 ADDITIONAL ABLATIONS

We provide more ablation experiments to demonstrate the representational benefits of DEQ-MPC.

**Representational hardness.** We look at the effect of increasing horizon length as it serves as a good proxy for various metrics such as problem conditioning, dimensionality, practical utility etc. Specifically, figure 8 shows the validation errors obtained by each model after training as we vary the horizon length from $T = 3$ to $T = 12$. We observe that the gap between the validation errors of the iterative models and the non-iterative ones is preserved even as we increase the size of the problem. Further, we observe that the representational benefits of the DEQ network in DEQ-MPC-DEQ starts becoming more obvious in the longer horizon problems as the difference in validation error between DEQ-MPC-DEQ and DEQ-MPC-NN increases. This illustrates the effectiveness of the infinite depth in DEQs helping with capturing the longer context.

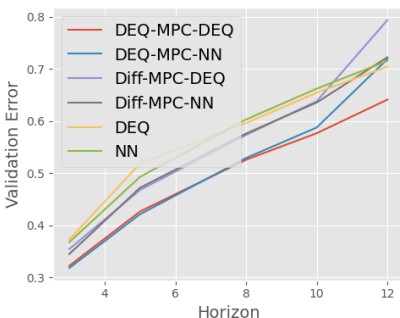

Figure 8: Time horizon ablations

**Validation error with iteration count.** Figure 9 shows the validation error across the Augmented Lagrangian iterations. As discussed earlier, the Diff-MPC variants here use the same predicted parameters throughout iterations while the DEQ-MPC variants use ADMM and thus update the optimization parameters using the network inference every two AL iterations. Interestingly, the gap in validation error starts accruing from the early AL iterations itself. But gap gets pronounced after the fourth AL iteration as the Diff-MPC variants saturate while DEQ-MPC continues to improve thanks to the repeated updates to the problem parameters.

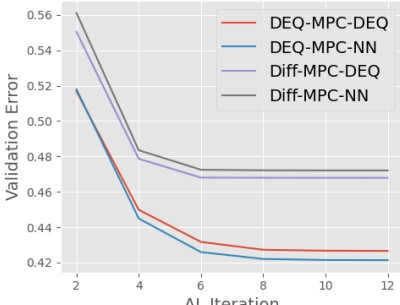

Figure 9: AL iteration ablations

### A.2 DEQ NETWORK GRADIENTS

Computing gradients through the fixed point iteration in a DEQ model typically requires using the implicit function theorem equation 2, which involves computing a linear system solve. However, recent work (Geng et al., 2021; Fung et al., 2021) has shown that the approximations of the gradient by simply assuming an identity Jacobian or differentiating through the last few iterations of the fixed point iteration using vanilla backpropagation is equally/more effective while being computationally cheaper. We adopt this approach. Specifically, we run the function a couple more times after computing the fixed point, and simply backpropagate through those last couple of iterations to compute the parameter gradients.

### A.3 AUGMENTED LAGRANGIAN ALGORITHM

Specifically, given the general MPC problem in equation 1, we form the following Lagrangian

$$\mathcal{L}(\tau, \lambda, \eta, \mu) = \sum_t C_{\theta,t}(\tau_t) + \lambda^T h_\theta(\tau) + \eta^T k_\theta(\tau_t, x_{t+1}) + \frac{\mu}{2}\|h_\theta(\tau_t)^+\|_2^2 + \frac{\mu}{2}\|k_\theta(\tau_t, x_{t+1})\|_2^2, \quad (16)$$

where $h_\theta(\tau_t) \leq 0$ are the inequality constraints and $k_\theta(\tau_t, x_{t+1}) = 0$ are all the equality constraints (including the dynamics and initial state constraints), $\lambda$ and $\eta$ are the corresponding Lagrange multipliers and $\mu > 0$ is the penalty parameter. $h_\theta(\tau_t)^+$ represents an element-wise clipping at zero $\max(0, h_\theta(\tau_t))$. The augmented Lagrangian method then proceeds by alternating between updating the primal variables ($\tau$), dual variables ($\lambda, \eta$) and the penalty parameter ($\mu$) as shown in algorithm 1.

---

**Algorithm 1** Augmented Lagrangian Solver for MPC-m

---

**Require:** Initialize $\tau^0$, $\lambda^0$, $\eta^0$, $\mu^0$ (warm-started using previous DEQ-MPC-iteration, parameters), $\gamma > 1$
 1: Set $j = 0$
 2: **repeat**
 3:    **Primal update:** Solve the unconstrained minimization problem using the Gauss-Newton method

$$\tau^{j+1} = \arg\min_{\tau} \mathcal{L}(\tau, \lambda^j, \eta^j, \mu^j)$$

 4:    **Dual update:** Update the Lagrange multipliers

$$\lambda^{j+1} = \max(\lambda^j + \mu^j h_\theta(\tau^{j+1}), 0)$$

$$\eta^{j+1} = \eta^j + \mu^j k_\theta(\tau^{j+1})$$

 5:    **Penalty update:** Update the penalty parameter

$$\mu^{j+1} = \gamma\mu^j$$

 6:    $j = j + 1$
 7: **until** Stopping criterion is met (or $j = m$ iterations)
 8: **return** Final solution $\tau^m, \lambda^m, \eta^m, \mu^m$

---

### A.4 NETWORK ARCHITECTURE DETAILS

We provide the details on the network architecture of the DEQ model and the feedforward network.

**Inputs.** For the DEQ-MPC variants, we have 2 inputs: ($x_0$, the initial state and $x_{1:T}^i$, the current state estimates from the optimizer). For the Diff-MPC variants, we only get $x_0$ as input. But we repeat it T times and concatenate it ($[x_0] * T$ to obtain a temporal input that can be fed to the temporal convolutional network described below.

**Feedforward network.** We use a Temporal Convolution Network architecture. We first compute input embeddings for the trajectory by computing a node embedding at each time-step with a node encoder (Linear-LayerNorm-ReLU). We then concatenate the node embedding of $x_0$ to all time-steps and a corresponding time embedding to indicate their respective time-steps. These are then passed through a series of four temporal convolution residual blocks (Conv1D-GroupNorm-ReLU) before computing the output with a final temporal Conv1D layer. The output at each time step represents the $\delta x_t = x_t - x_0$. More details are available in the code attached with the supplementary.

**DEQ network.** We again use a Temporal Convolution Network architecture. We have three separate blocks here, namely, input injection layer $I$, fixed point layer $d$ and output layer $g$. We compute the forward pass by first computing the fixed point on the latents:

$$z^* = d_\phi(z^*, I_\phi(x_0, \hat{x}_{1:T})) \tag{17}$$

and then compute the outputs using $g_\phi(z^*)$. The input injection layer is similar to the feedforward network. We compute a node embedding at each time-step with a node encoder (Linear-LayerNorm-ReLU) and then concatenate the node embedding of $x_0$ and a corresponding time embedding to all time-step node embeddings. This sequence of concatenated node embeddings are then passed through a TCN block (Conv1D-GroupNorm-ReLU) to get the final input embeddings that are fed to the fixed point layer.

The fixed point layer : The input embeddings are passed through a TCN block (Conv1D-GroupNorm-ReLU) and added to a temporally arranged latent variable $z$. The resulting embeddings are passed through another TCN block (Conv1D-GroupNorm-ReLU) with a residual connection, to obtain the output $z$. These operations combined represent $d_\phi$. We compute the fixed point of this layer using a standard Anderson acceleration fixed point solver (Anderson, 1965; Walker & Ni, 2011) to get the resulting $z^*$.

The output layer $g_\phi(z^*)$ is again a TCN block (Conv1D-GroupNorm-ReLU-Conv1D) that computes the computes the output $\delta x_t = x_t - x_0$.

More details are available in the code attached with the supplementary.

**Default hyperparameters** We use a hidden size of 256 for the Pendulum, 512 for Cartpole, 512 for Quadrotor, 1024 for QPole and QPoleObs unless otherwise specified. During training, we use a batch size of 200 for all environments.

### A.5 NOTE ON CONVERGENCE

Our treatment of the joint system as a DEQ allows us to borrow results from Winston & Kolter (2020)Bai et al. (2021) to ensure convergence of the fixed point iteration. Specifically, if we assume the joint Jacobian of the ADMM fixed point iteration is strongly monotone with smoothness parameter $m$ and Lipschitz constant $L$, then by standard arguments (see e.g., Section 5.1 of (Ryu & Boyd, 2016)), the fixed point iteration with step size $\alpha < m/L^2$ will converge. However, going from the strong monotonicity assumption on the joint fixed point iterations to specific assumptions on the network or the optimization problem is less straightforward. But, in practice a wide suite of techniques have been used to ensure that such fixed points exist and can be found using relatively few fixed point iterations. In fact, for all of our experiments, we converge within 6 ADMM iterations once trained.

