# OpenReview forum: "DEQ-MPC : Deep Equilibrium Model Predictive Control"
_ICLR.cc/2025/Conference — Submitted to ICLR 2025_

### Official Review · Reviewer_jwqW · 2024-10-29

**Soundness:** 1
**Presentation:** 1
**Contribution:** 2
**Rating:** 3
**Confidence:** 5

**Summary:**

The paper introduces an approach to integrating MPC with deep learning by combining optimization solvers and network inference in a unified fixed-point framework. The proposed DEQ-MPC iteratively alternates between deep equilibrium models neural network predictions and MPC optimizations using the ADMM method until a stable equilibrium is reached.

**Strengths:**

The paper's strength is its approach of updating the neural network and MPC simultaneously using ADMM, allowing adaptive, joint optimization that improves stability, gradient alignment, and efficiency.

**Weaknesses:**

The paper presents an interesting approach; however, there are some areas for potential improvement. First, it lacks a novel theoretical contribution and does not provide formal proofs to support its framework. Additionally, there is minimal analysis of computational expense, which would strengthen the understanding of its practical feasibility. The presentation could also benefit from greater clarity, as certain aspects, such as the representation of parameter theta, are not entirely clear, which may make the paper challenging to follow. While the paper mentions task-specific priors, concrete examples or integration of these are not demonstrated, which could further enhance the practical relevance of the approach.

**Questions:**

1. Could the authors elaborate on any theoretical foundations or formal proofs that validate the convergence of the proposed framework?
2. What is the computational overhead of DEQ-MPC compared to traditional differentiable MPC? Have there been any benchmarks or evaluations on computational efficiency, especially in real-time applications?
3. The paper mentions the use of parameter theta, but its role and significance are unclear. Could the authors clarify what theta represents?
4. Although the paper claims to incorporate task-specific priors, there are no explicit examples. Could the authors provide examples of how task-specific priors are integrated into DEQ-MPC, and the impact these have on task performance?

---

> ### Author Response · Authors · 2024-11-28
> **Authors response**
>
> We thank the reviewer for their detailed feedback. We’re glad the reviewer appreciated the novelties in the method. We discuss some of the criticisms and questions pointed out by the reviewer as follows :
> 1. **Theoretical Contribution** :
>     - **Note on Convergence** : We have added a note around convergence analysis in the appendix when discussing the solver details.
> Our treatment of the system as a DEQ allows us to borrow results from [1][2] to ensure convergence of the fixed point iteration. Specifically, if we assume the joint Jacobian of the fixed point iterations we describe are strongly monotone with smoothness parameter $m$ and Lipschitz constant $L$, then by standard arguments (see e.g., Section 5.1 of [3]), the fixed point iteration with step size $\alpha<m/L^2$ will converge. However, going from the strong monotonicity assumption on the joint fixed point iterations to specific assumptions on the network or the optimization problem is less straightforward. But, in practice a wide suite of techniques have been used to ensure that such fixed points exist and can be found using relatively few fixed point iterations. In fact, for all of our experiments, we converge within 6 ADMM iterations once trained.
>     - **Representational Benefits** : Perhaps a more central claim in the paper is around the representational benefits of our approach over vanilla differentiable MPC. However, given the non-linear nature of the networks and optimization problems we consider, it’s unclear to us how we could theoretically demonstrate the representational benefits of the approach beyond the intuitive reasoning we provide in the paper. We believe this would be a significant theoretical undertaking and hope that future work will address these representational benefits more rigorously.
> 	Broadly, we view the main contribution of our paper as providing the algorithmic tools and empirical results and ablations to leverage these representational benefits when combining optimization layers within deep learning pipelines in a principled way.
> 2. **Computational expense** : For most of the experiments, we compare the methods in the streaming setup with warm-starting, i.e 1 ADMM iteration for DEQ-MPC methods vs a network inference + 2 AL iterations for the Diff-MPC methods. In this setting, the two methods have almost exactly the same runtime (most of it consumed in the optimization iterations) : 0.121s vs 0.120s.
> On the other hand, when used without warm-starting, the run-time differences between the two methods are non-insignificant : runtimes of 1.12s and 1.18s for DEQ-MPC-NN and DIFF-MPC-NN respectively without any warm-starting. This is despite the fact that we run more network inference steps (6 vs 1) and more total AL iterations (12 vs 10) for DEQ-MPC. We observe that DEQ-MPC requires fewer line-searches compared to Diff-MPC and each inner AL solve converges faster given the network adapts the references across iterations. This was interesting given this was never explicitly incentivized in the network learning objectives and was learnt automatically by DEQ-MPC.
> However, we do not report these runtime results in the paper given the optimization solver we implemented is in pytorch and hence not optimized for real time usage. Thus, quantitative runtime comparisons are somewhat misleading.
> 3. **Writing Clarity** : Pardon us for the confusion. We try to ground these variables more concretely in section 4.1 in the context of trajectory tracking. The optimization parameter in that context for example would represent the waypoints we expect the robot to follow and the final optimized trajectories are represented as $\tau = [x,u]$. Do let us know if the description there was unclear.
> 4. **Task specific priors** : Pardon us again for the confusion here. But in the context of Differentiable-MPC and DEQ-MPC, we refer to the incorporation of the optimization problem as part of the prediction problem itself as the priors as opposed to predicting the trajectories directly as outputs of a network. Thus, we are able to explicitly incorporate objectives and constraints (eg. dynamics, control limits, obstacle avoidance constraints) to bake in specific properties of the system we might know as part of the end-to-end prediction problem. We show this across examples.
>
> [1] Ezra Winston and J Zico Kolter. Monotone operator equilibrium networks. Advances in neural information processing systems, 2020.
>
> [2] Shaojie Bai, Vladlen Koltun, and Zico Kolter. Stabilizing equilibrium models by jacobian regularization. In Proceedings of the International Conference on Machine Learning, 2021.
>
> [3] Ernest K Ryu and Stephen Boyd. Primer on monotone operator methods. Appl. Comput. Math, 2016.

---

> > ### Author Response · Authors · 2024-12-02
> > **Gentle reminder**
> >
> > We hope our rebuttal/responses during the discussion period, and the changes made to the draft have addressed the questions and weaknesses in your reviews. We are happy to address further questions or concerns about our contributions!

---

### Official Review · Reviewer_WgeY · 2024-10-29

**Soundness:** 3
**Presentation:** 3
**Contribution:** 3
**Rating:** 6
**Confidence:** 3

**Summary:**

The paper presents a novel deep learning method integrating dynamics learning and model predictive control (MPC). Optimization is performed in two stages: update weights then solve MPC repeat. This yields several advantages once the model is trained. Examples of these are smoother gradients during training and for MPC. The model can be warm-started much more readily than alternative methods and constraints can be strictly adhered to also. The authors demonstrate the effectiveness of their methods using five examples and compare their algorithms to alternative Diff-MPC methods.

**Strengths:**

The strengths of this paper are numerous. The algorithmic setup is original and an interesting idea. The contributions are compelling.

The authors show that this method outperforms alternative Diff-MPC methods in the environment setups. The algorithm is well-ablated and compelling enough with minor changes to justify the authors’ claims. The paper is well-presented and easy to follow. Though I do not know the Diff-MPC literature well, this paper appears significant enough for publication.

**Weaknesses:**

Despite the numerous strengths, there are a few weaknesses. However, these are readily addressable.

The majority of the weaknesses revolve around the reporting of the results. There are no standard deviation error bars on the reported numbers or figures (Fig. 2 to 7) or Table 1. Stating the number of experiments completed and adding standard deviation as a plus/minus spread of error in Table 1 would strengthen the paper. Comparing the statistical significance of the reported rewards in Table 1 would improve the paper also. For example, it would be interesting to know if there was any statistical difference in performance between DEQ-MPC-NN and DEQ-MPC-DEQ in the Quad-Pole scenario. A Mann-Whitney U-test could work here. I leave it up to the authors to choose how they evaluate their statistics. Similarly, figures Fig. 2 to 7 do not show the standard deviations of the validation error or the normalised returns  across multiple training runs. This would strengthen the comparison.

**Questions:**

As seen in the weaknesses section, the reporting of results is a limitation of the paper. However, this is easily remedied. Please follow my recommendations in the weaknesses section. Please justify the statistical comparisons you use. Other than this, the paper is well-written, original, and of significance.

---

> ### Author Response · Authors · 2024-11-28
> **Authors response**
>
> We thank the reviewer for their thoughtful feedback. We’re glad the reviewer appreciated the novelties in the method, the diverse ablations, empirical results and clear writing. We acknowledge the missing error bars in the original submission. We have now revised the paper with the provided feedback. We have included mean and variance of the normalized returns on all the experiments in the table and have also provided the error bars for the ablation experiments. We thank the reviewer for their feedback!

---

> > ### Author Response · Authors · 2024-12-02
> > **Gentle reminder**
> >
> > We hope our rebuttal/responses during the discussion period, and the changes made to the draft have addressed the questions and weaknesses in your reviews. We are happy to address further questions or concerns about our contributions!

---

### Official Review · Reviewer_pXdj · 2024-10-30

**Soundness:** 3
**Presentation:** 3
**Contribution:** 3
**Rating:** 6
**Confidence:** 4

**Summary:**

This paper considers differentiable model predictive controllers with parameters that depend on the output of a neural network. Standard practice is to run the network once and then solve the MPC optimization problem. However, the authors argue that this may yield optimization problems which are challenging to solve, potentially leading to unstable training dynamics. Instead, they propose to condition the neural network on the solver state and alternate between running the network and solving the optimization problem. They also explore making the network a deep equilibrium model, which is an implicit layer that converges to a fixed point. Therefore, both network inference and the optimzation solver become iterative algorithms, which can alternate until convergence. The authors evaluate their proposed approach, DEQ-MPC, on a number of simulated dynamical systems engaged in a waypoint tracking task, in which the waypoints are the output of the neural network. They compare to Diff-MPC, which uses the exact same solver as their method, except they only run the neural network once prior to solving the MPC optimization problem. Their evaluations show that DEQ-MPC outperforms Diff-MPC on all benchmark systems. Moreover, through extensive evaluations, they find that the deep equilibrium model generally outperforms using a standard neural network. They also explore benefits of DEQ-MPC in terms of generalization, network capacity, sensitivity to constraints, validation loss curve monotonicity, sensitivity to cost function parameters, and warm-starting.

**Strengths:**

- Improving the performance of structured policy classes, such as differentiable MPC, is an important and timely problem.
- The paper is well organized and overall clearly written. It does a good job explaining the novelty and results and provides enough information to support its claims.
- The results are promising and indicate that DEQ-MPC can outperform more standard implementations of differentiable MPC by alternating between network inference and the optimization solver. Although the tasks considered are all trajectory tracking, they provide extensive evaluation of their method compared to baselines and tease apart what aspects are most important.

**Weaknesses:**

- It would be great if Table 1 included end-to-end neural network results as well. This is an important comparison point that is discussed a bit in the ablations. But the paper would benefit from highlighting this more centrally.
- There is a lot of discussion about gradient smoothness and alignment with the global or desired update direction. However, the analysis really only looks on loss curves. Given that this is discussed so heavily in the argument for the method, it would be great to see how well the IFT gradients align with the true gradients that would be computed via backpropagation. If not that, at least look at the smoothness of gradients across epochs during training.
- A minor point, but I don't feel that Figure 1 conveys enough information about the approach. It would be nice if the DEQ-MPC layer part highlighted that the network inference could also be iterative, rather than just conditioned on solver state.
- The evaluations only really consider one task, which is waypoint tracking. Although this is evaluated across many different systems, it would really strengthen the paper to consider other flavors of tasks as well. Or maybe even inferring dynamics parameters too, rather than just terms in the cost function.
- The evaluations also only consider fairly short-horizon tasks (T=5). It would be great to see how results scale with longer horizons. And same goes for evaluating the warm-starting ability of DEQ-MPC. There are very few warm-starting steps during training and evaluation.
- There are some details that appear to be missing (unless I missed them), such as the number of fixed point iterations for the DEQ used within DEQ-MPC.
- DEQ-MPC appears to also use intermediate losses that compute gradients through multiple iterates, rather than just the final solution. It is unclear if diff-MPC is also trained this way. If not, it would be an important ablation to see if it would help improve diff-MPC's performance.

**Questions:**

- Are all cost function and dynamics model parameters manually set or also learned end-to-end? If manually set, how would this approach extend to the scenario where we want to learn dynamics, cost, or both jointly in an end-to-end fashion?
- How many iterations are used running each DEQ model, or are they run until a fixed-point is reached?
- Does the Diff-MPC baseline use the augmented Lagrangian method or is it iLQR like in the original paper? Does the network it use also contain the temporal convolutions?
- Would Diff-MPC perform better if its gradients were computed using multiple intermediate iterates, like in the proposed DEQ-MPC approach? This was one aspect of the ablation that appeared to be missing.
- Are the IFT gradients only valid at a fixed point? If so, using them to compute gradients on intermediate solutions that have not converged may give incorrect, although still useful, gradients. How do the gradients computed through the IFT compare to the exact gradients computed via autodiff?

---

> ### Author Response · Authors · 2024-11-28
> **Authors response**
>
> We thank the reviewer for their detailed feedback. We’re glad the reviewer appreciated the importance of the problem, the novelties in the method, the diverse ablations, empirical results and clear writing. We address the specific questions and weaknesses pointed out by the reviewer.
> 1. **End-to-end neural net baselines** : The end-to-end neural net baselines in the ablations represent simply measuring the validation error on the waypoints predicted by the network. We do not include the end-to-end neural net baselines in the table given that tracking those waypoints post-hoc using the mpc solver in a warm-started setting typically just barfs and gets unstable in a number of cases.
> 2. **Gradient smoothness/alignment** : We believe the discussion of global/desired update direction was a bit of an overreach and have edited it in the draft to focus on smoothness instead.
> 3. **Figure 1 corrections** : We acknowledge that adding an iterative component to the network architecture in Figure 1 might make it more informative. However, we believe that it somewhat distracts from the main point of the paper which is that DEQ-MPC layers benefit representationally from conditioning the network on the optimization iterates. The additional occasional stability/representational benefits from making the network itself iterative (i.e DEQ) was almost an afterthought from the perspective of the paper. Thus we chose to not make it explicit in the figure either.
> 4. **Predicting dynamics/constraint parameters** : We acknowledge that in our current experiments we formulate the NN output / MPC parameters as waypoints and the MPC objective as waypoint tracking. We agree that our work can be strengthened by wiring those modules differently such as learning the dynamics or constraints (obstacle positions). Indeed we hope to explore those and other tasks in our future work. However, we focus specifically on the trajectory tracking task in this paper given its wide ranging applications. That helps us ground the method in an important task while keeping the formulation consistent and simple across experiments. This helps us focus specifically on the modelling and methodological choices of our work.
> 5. **Effect of number of warm-starting steps** : We show some ablation experiments for long-horizon evaluations in the appendix Figure 8. Regarding the warm-starting steps, for our experiments, we tested with 2, 4 and 6 warm-starting AL iterations and 1, 2, 3 ADMM iterations respectively in Figure 7. We found that even with just 1 ADMM iteration, DEQ-MPC performs pretty well. In fact, as we increase to 6 warm-started AL iterations, even the Diff-MPC variants start performing competitively.
> 6. **Convergence criterion for DEQ** : Pardon us for the missing detail. We have added it to the architectural details in the appendix. For the DEQ network, we solve it to convergence using a fixed point solver (to a tolerance of 1e-2). For the outer DEQ, we use 6 ADMM iterations across experiments when not warm-starting.
> 7. **Intermediate gradients for Diff-MPC** : Using intermediate gradients for Diff-MPC biases the network gradients with the earlier iteration gradients. Given that the network in this case can’t distinguish between ite	rations (due to the lack of iterate conditioning), this results in worse learning. Whereas, given that the network in DEQ-MPC is conditioned on the optimization iterate, using intermediate iteration gradients for DEQ-MPC does not lead to any biases.
> In the trajectory tracking problem we study, we learn the waypoints for the optimization problem, which appear as cost parameters. Other cost and dynamics parameters are indeed set manually currently. While they could conceivably be learnt jointly using gradient descent, we believe it might be unstable to train especially when the parameters are wildly off. Thus, we believe that, in such a scenario, it might be a good idea to first fit the cost and dynamics parameters independently using a proxy task. Example, for the dynamics, one could just fit the dynamics parameters by minimizing the dynamics residuals on the data provided or directly on the trajectory tracking problem using IFT gradients. Likewise, the cost terms could be learnt by using perturbed tracking trajectories and then learning the cost terms on the corresponding end-to-end tracking problem using IFT gradients.

---

> > ### Author Response · Authors · 2024-11-28
> > **Authors response continued**
> >
> > 8. **Architecture details** : All networks use temporal convolutions and are similarly parameterized.
> > 9. **Solver details** : We use the AL solver for the Diff-MPC variants as well given that the iLQR implementation from the original Diff-MPC paper does not conveniently handle state constraints or other inequality constraints other than control limits.
> > 10. **IFT gradient validity** : The IFT gradients are defined assuming the network reached a fixed point. But what they are effectively doing intuitively is multiplying the inverse of the KKT matrix of the optimization problem linearized at the current iterate to the gradient. So even when the fixed point is not converged, using the IFT gradients effectively ‘informs’ the gradients about the local landscape of the problem. In fact, this is exactly what an autodiff library would implement to compute the gradients even for unconverged iterates of a fixed point iteration.

---

> > > ### Author Response · Authors · 2024-12-02
> > > **Gentle reminder**
> > >
> > > We hope our rebuttal/responses during the discussion period, and the changes made to the draft have addressed the questions and weaknesses in your reviews. We are happy to address further questions or concerns about our contributions!

---

### Official Review · Reviewer_7Dkz · 2024-11-03

**Soundness:** 3
**Presentation:** 3
**Contribution:** 2
**Rating:** 3
**Confidence:** 4

**Summary:**

Incorporating task-specific priors like auxiliary constraints in policy training for robotic control can improve safety, flexibility, and generalization. Differentiable Model Predictive Control layers allow such constraints to be embedded directly in neural networks, enabling end-to-end training while retaining interpretability. However, standard differentiable MPC treats solvers as black-box layers, leading to potential instability and inefficiencies. To address this, the paper introduces Deep Equilibrium Model Predictive Control (DEQ-MPC)  to unify the optimization solver and network. The enables a joint inference-optimization approach that improves gradient flow, warm-starting, and stability in complex tasks. DEQ-MPC performs well in warm-starting scenarios. It has reduced iteration needs and this is useful for real-world deployment. The paper introduces two variants: DEQ-MPC-NN, and DEQ-MPC-DEQ. These variants highlight the trade-offs between performance and stability. The authors suggest that DEQ-MPC could be expanded to reinforcement learning and broader constrained optimization problems in the future.

**Strengths:**

The network and optimization solver unification enhances gradient flow and produces more stable and efficient training dynamics compared to traditional methods.

DEQ-MPC allows for seamless integration of constraints. This in turn gives better safety and reliability, which are important for safety-critical applications such as the robot applications.

DEQ-MPC reduces computation through efficient warm-starting

**Weaknesses:**

The joint inference-optimization structure is more complex to implement  and deploy.

Current evaluations focus on toy examples like pendulum, cart pole, and trajectory tracking, so DEQ-MPC’s effectiveness is not clear. We do not need machine learning for pendulum and cart pole.  As such, the paper results are preliminary.

**Questions:**

The evaluation of the method is weak. Pendulum and cart pole are useful for debugging a method but not for arguing its effectiveness. Can you evaluate your method on more compelling cases? Unless we see some significant results for more realistic problems this solution remains a theoretical contribution.

Can you comment on the complexity of the solution -- performance, energy requirements, and implementation requirements.

Can you explain more clearly and comprehensively how you wire this architecture?

Can you summarize the properties of this approach? How large are the models? How do you train them? What is the cost of training? What is the cost of inference?

---

> ### Author Response · Authors · 2024-11-28
> **Authors response**
>
> We thank the reviewer for their detailed feedback. We’re glad the reviewer appreciates the importance of the underlying problem we seek to solve and the nuances of the methodological contributions. We address the specific questions and weaknesses pointed out by the reviewer.
> 1. **Complicated approach** :  At a fundamental level, the methodology is motivated by the fact that for representational effectiveness of learning+optimization layers, we desire a coupled inference problem where the network inference is a function of the optimization iterates (as observed in our representational ablations). We believe that’s a fundamental representational unlock needed to exploit the implicit/iterative nature of optimization problems which hasn’t been explored in previous work. The specific algorithmic contributions around the scheduled alternating optimization etc, are simply a means to achieve this. They allow us to solve the alternating problem in an efficient manner. But we believe those are important details towards making these approaches fast enough for real-time applications!
> 2. **Weak evaluation** : As pointed out by the reviewer we indeed use pendulum/cartpole for analysis and debugging purposes (e.g gradient analysis). But we also include extensive experiments on more complicated problems such as
>
>     (a) Quadrotor goal reaching task : We use the full nonlinear dynamics of the quadrotor with motor thrust actions (considering the agile low-level dynamics), unlike many other work which output thrust and velocities, abstracting what we consider in our work.
>
>     (b) Quadrotor-pole :  where a pole is attached to the quadrotor and the task is to swing up the pole (using its inertia) while reaching the goal. This task is dynamic and challenging, highlighting the importance of incorporating the MPC layer.
>
>     (c) Quadrotor-pole with obstacles : which involves solving the above problem while avoiding obstacles in the path. This task is not just a simple 3D navigation problem as it features high complexity and nonconvexity of the dynamics and obstacle avoidance. This task is also considered safety-critical, as the policy must reason about both hard input and state constraints.
> Moreover, we run all the ablation experiments on (b) and (c) which we consider fairly complicated environments as observed from the results. Through these ablations, we tease out the specific representational benefits obtained from this approach. While we acknowledge that more complicated tasks could also have been tried, we believe these experiments demonstrate the representational benefits that could be gained from our method in fairly complicated situations as well.
>
> 3) **Architectural and solver details** : We provide some of the solver and architecture details in the appendix in section A.3 and A.4 and provide the code for any additional nuances we may have missed in our description. We would be happy to provide any additional details if the reviewer has specific questions in that regard!
>
> 4) **Computational expense** : For most of the experiments, we compare the methods in the streaming setup with warm-starting, i.e 1 ADMM iteration for DEQ-MPC methods vs a network inference + 2 AL iterations for the Diff-MPC methods. In this setting, the two methods have almost exactly the same runtime (most of it consumed in the optimization iterations) : 0.121s vs 0.120s.
> On the other hand, when used without warm-starting, the run-time differences between the two methods are non-insignificant : runtimes of 1.12s and 1.18s for DEQ-MPC-NN and DIFF-MPC-NN respectively without any warm-starting. This is despite the fact that we run more network inference steps (6 vs 1) and more total AL iterations (12 vs 10) for DEQ-MPC. We observe that DEQ-MPC requires fewer line-searches compared to Diff-MPC and each inner AL solve converges faster given the network adapts the references across iterations. This was interesting given this was never explicitly incentivized in the network learning objectives and was learnt automatically by DEQ-MPC.
> However, we do not report these runtime results in the paper given the optimization solver we implemented is in pytorch and hence not optimized for real time usage. Thus, quantitative runtime comparisons are somewhat misleading.

---

> > ### Author Response · Authors · 2024-12-02
> > **Gentle reminder**
> >
> > We hope our rebuttal/responses during the discussion period, and the changes made to the draft have addressed the questions and weaknesses in your reviews. We are happy to address further questions or concerns about our contributions!

---

### Official Review · Reviewer_9MzT · 2024-11-06

**Soundness:** 3
**Presentation:** 4
**Contribution:** 2
**Rating:** 5
**Confidence:** 4

**Summary:**

This paper introduces DEQ-MPC, a framework that alternates optimizing the network and differentiable MPC layers like a deep equilibrium model. By jointly optimizing network and MPC solver states as a fixed-point problem, DEQ-MPC achieves smoother gradients, better warm-starting, and improved performance. The results are demonstrated in imitation learning on several classical robotic control tasks.

**Strengths:**

1. The authors propose coupled dynamics and MPC layer updates instead of decoupled updates in Diff-MPC, which offers better empirical performance, smoother gradients and leverages warm-starting. The combination of DEQ and Diff-MPC is novel.
2. The authors conduct diverse ablation studies to prove the effectiveness of the design choices.
3. The paper writing is clear and intuitive to read.

**Weaknesses:**

1. The technical methodology is incremental and not very inspiring. Though it provides improved performance compared to Diff-MPC, it complicates the approach by scheduling the alternating optimization and more hyperparameters.
2. The experimental validations are done with a single random seed, which has the risk of overfitting. I suggest the authors try generating different versions of the dataset with different seeds or at least try different random partitions of the dataset. Then, report the performance with mean and variance.
3. No theoretical insights into why the proposed approach works better.

**Questions:**

1. What are the convergence criteria used for computing validation errors of all methods?
2. What’s the runtime of the method compared to Diff-MPC?
3. How can this approach be practically more useful for RL or high-dimensional tasks?
4. Fig 1 does not look intuitive - what do the drones and bounded curves mean?

---

> ### Author Response · Authors · 2024-11-28
> **Authors response**
>
> We thank the reviewer for their detailed feedback. We’re glad the reviewer liked the diverse ablations, empirical results and clear writing. We address the specific questions and weaknesses pointed out by the reviewer.
> 1. **Incremental methodology+complicated approach** : At a fundamental level, the methodology is motivated by the fact that for representational effectiveness of learning+optimization layers, we desire a coupled inference problem where the network inference is a function of the optimization iterates (as observed in our representational ablations). We believe that’s a fundamental representational unlock needed to exploit the implicit/iterative nature of optimization problems. As shown in our experiments, this improves the generalization and scaling abilities of the model. But more importantly it allows us to utilize warm-starting more effectively which was not possible with traditional differentiable mpc methods. The specific algorithmic contributions around the scheduled alternating optimization etc, are simply a means to achieve this. They allow us to solve the alternating problem in an efficient manner.
> 2. **Reporting error bars** : We updated all our experiments/ablations as suggested by the reviewer and ran multiple runs with different dataset partitions and reported the mean and variance for all the results and ablations.
> 3. **Theoretical insights** : Our central claim in the paper is around the representational benefits of our approach over vanilla differentiable MPC. However, given the non-linear nature of the networks and optimization problems we consider, it’s unclear to us how we could theoretically demonstrate the representational benefits of the approach beyond the intuitive reasoning we provide above. We believe this would be a significant theoretical undertaking and hope that future work will address these representational benefits more rigorously.
> 4. **Convergence criteria** : We run the methods in a non-streaming setup. For the DEQ-MPC variants we run the solve the joint problem for 6 ADMM iterations whereas for the Diff-MPC variants, we run the Augmented Lagrangian solver for 10 AL iterations. We observe that the solver in both cases converges to <1e-3 tolerance on the steps.
> 5. **Computational costs** : For most of the experiments, we compare the methods in the streaming setup with warm-starting, i.e 1 ADMM iteration for DEQ-MPC methods vs a network inference + 2 AL iterations for the Diff-MPC methods. In this setting, the two methods have almost exactly the same runtime (most of it consumed in the optimization iterations) : 0.121s vs 0.120s.
> On the other hand, when used without warm-starting, the run-time differences between the two methods are non-insignificant : runtimes of 1.12s and 1.18s for DEQ-MPC-NN and DIFF-MPC-NN respectively without any warm-starting. This is despite the fact that we run more network inference steps (6 vs 1) and more total AL iterations (12 vs 10) for DEQ-MPC, we observe that DEQ-MPC requires fewer line-searches compared to Diff-MPC and each inner AL solve converges faster given the network adapts the references across iterations. This was interesting given this was never explicitly incentivized in the network learning objectives and was learnt automatically by DEQ-MPC.
> However, we do not report these runtime results in the paper given the optimization solver we implemented is in pytorch and hence not optimized for real time usage. Thus, quantitative runtime comparisons are somewhat misleading.
> 6. **Utility for RL/high-dimensional tasks** : As discussed above, the main benefits of DEQ-MPC layers over DIFF-MPC layers is representational. Thus, we believe these methods would directly transfer to most other problems where differentiable-mpc layers have already been shown to be effective. Eg : [1] for RL or [2] in high dimensional partially observed environments.
> 7. **Figure 1 description** : Pardon us for the lack of clarity here. We have modified the figure description for additional clarity. The drone inputs were meant to show observations from the robot and the bounded curves are the waypoints with bounds representing constraints which are violated by \theta but satisfied by \tau.
>
> [1] Romero, Angel, Yunlong Song, and Davide Scaramuzza. "Actor-critic model predictive control.", IEEE International Conference on Robotics and Automation (ICRA). IEEE, 2024
>
> [2] Acerbo, Flavia Sofia, et al. "Learning from Visual Demonstrations through Differentiable Nonlinear MPC for Personalized Autonomous Driving." arXiv preprint arXiv:2403.15102 (2024).

---

> > ### Author Response · Authors · 2024-12-02
> > **Gentle reminder**
> >
> > We hope our rebuttal/responses during the discussion period, and the changes made to the draft have addressed the questions and weaknesses in your reviews. We are happy to address further questions or concerns about our contributions!

---

### Meta-Review · Area_Chair_PUio · 2024-12-20

**Metareview:**

The paper introduces DEQ-MPC, a novel method that integrates deep learning and model predictive control (MPC) by alternating optimizing the differentiable MPC layers with the network in a manner similar to DEQs. This approach, unlike standard differentiable MPC, couples the network and solver, and leads to improvements in gradient stability and smoothness, warm-starting, and generalization. DEQ-MPC is evaluated on simulated robotic tasks, demonstrating superior performance over Diff-MPC, which uses the same solver but only runs the deep network once before solving the MPC problem (instead of alternating between the two). Two variants are explored: DEQ-MPC-NN and DEQ-MPC-DEQ.

DEQ-MPC's key strength lies in the originality of its idea, which may not be earth-shatteringly innovative, but leads to smoother gradients, more stable training, and efficient warm-starting. Reviewers also praised the thorough ablations and clear presentation, and the authors addressed several reviewer questions during the rebuttal period by adding error bars and providing computational runtime comparisons.

The main weakness lies in the evaluation of the approach. As noted by several reviewers, there are no theoretical guarantees presented in this paper, so the evaluation comes to the empirical validation of the method. Several reviewers noted concerns about the relative simplicity of the environments -- the most complex involves a state dimension of 14 and control dimension of 4. The authors note that since DEQ-MPC is better than Diff-MPC while requiring no additional information relative to Diff-MPC, it should improve in any setting where Diff-MPC is currently used. However, given the lack of theoretical foundation in the paper, it seems critical to demonstrate this practically in the paper in order for it to be impactful in the wider community.

This paper has a lot of potential, but the reviewers agreed that it could be made stronger by evaluating on more challenging environments which are now the norm, given that no theoretical formalism is provided. This is a borderline paper, but in its current form, I recommend rejection.

**Additional Comments On Reviewer Discussion:**

Multiple reviewers asked for error bars (or confidence intervals, or variances) to be included in the reporting of the results. The authors addressed this by adding these to the revised manuscript (in Table 1), and it is evident with these error bars that there is a significant difference in performance for at least 4 / 5 of the tested environments.

Multiple reviewers also asked for more complex environments to be experimented with, but this was not done for the revision.

Several reviewers brought up questions about computational complexity / runtime, which were answered by the authors during the rebuttal. The runtime of DEQ-MPC is comparable to or better than Diff-MPC.

---

### Decision · Program_Chairs · 2025-01-22

Reject